# Machine learning predicts lifespan and suggests underlying causes of death in aging *C. elegans*

Carina C. Kern[1,2], Petru Manescu[3], Matt Cuffaro[1], Catherine Au [1], Aihan Zhang[1], Hongyuan Wang[1], Ann F. Gilliat[1], Sophie van Schelt[1], Marina Ezcurra [1,4,5] ✉ & David Gems [1,5]

Aging leads to age-related pathology that causes death, and genes affect lifespan by determining such pathology. Here we investigate how age-related pathology mediates the effect of genetic and environmental interventions on lifespan in *C. elegans* by means of a data-driven approach employing machine learning (ML). To this end, extensive data on how diverse determinants of lifespan (sex, nutrition, genotype, mean lifespan range: 7.5 to 40 days) affect patterns of age-related pathology was gathered. This revealed that different life-extending treatments result in distinct patterns of suppression of senescent pathology. By analysing the differential effects on mid-life pathology levels and lifespan, the ML models developed were able to predict lifespan variation, explaining 79% of the variance. Levels of pathology in the pharynx and intestine proved to be the strongest predictors of lifespan. This suggests that elderly *C. elegans* die predominantly from late-life disease affecting these organs. In addition, we noted profound sex differences in age-related pathology: the striking age-related pathologies in hermaphrodites affecting organs linked to reproduction are absent from males, suggesting that reproductive death may be hermaphrodite limited.

Aging (senescence) is the main cause of late-life disease, including Alzheimer's disease, many forms of cancer, cardiovascular disease, and type 2 diabetes mellitus. As such it is a major and increasing challenge to human well-being and to healthcare systems worldwide[1]. In animal models genetic, environmental and pharmacological interventions can increase lifespan and delay the onset of age-related diseases, and this has suggested possible mechanisms of aging[2–7]. However, it remains unclear whether such interventions are efficacious in humans, or how such mechanisms act as etiologies of late-life diseases.

The latter question is pressing in research on the nematode *Caenorhabditis elegans*. Here studies of long-lived mutants have identified signalling pathways with effects on lifespan that show evolutionary conservation across species[8]. Yet how such signalling pathways control senescent pathology remains unclear: our understanding of the causal chain of events from gene to signalling pathway to pathology to death remains incomplete (Fig. 1a). In this study we investigate how interventions altering lifespan affect pathology, and how pathology affects lifespan.

Some knowledge of *C. elegans* age-related pathology already exists thanks to several pioneering studies using light microscopy, automated image analysis and electron microscopy[9–14]. To expand this knowledge, we have systematically examined senescent pathology under a wider range of interventions that alter lifespan. Machine learning (ML) data-driven approaches were then applied, and ML models developed, to investigate how pathology affects lifespan. The decision to apply ML was driven by our mathematical and theoretical prediction that age-related pathologies are context-dependent. The impact of a given pathology, even at the same severity, may vary based on factors such as species, genotype and co-morbidities, even in a simple organism like *C. elegans*. Mathematically, rather than following linear or additive trends, lifespan is likely shaped by complex pathological interactions. Results support these predictions and the analysis provides new insights into the relationship between interventions, pathology and lifespan.

## Results

### Life-extending treatments differentially affect pathologies of aging

For ML analysis of the relationship between age-related pathology and lifespan, we first compiled a large dataset, spanning a wide range of interventions. This included both previously published data and data newly

[1]Institute of Healthy Ageing, and Research Department of Genetics, Evolution and Environment, University College London, London, UK. [2]LinkGevity, Babraham Research Campus, Cambridge, UK. [3]Department of Computer Science, Faculty of Engineering Sciences, University College London, London, UK. [4]School of Natural Sciences, University of Kent, Canterbury, UK. [5]These authors contributed equally: Marina Ezcurra, David Gems. ✉e-mail: m.ezcurra@kent.ac.uk

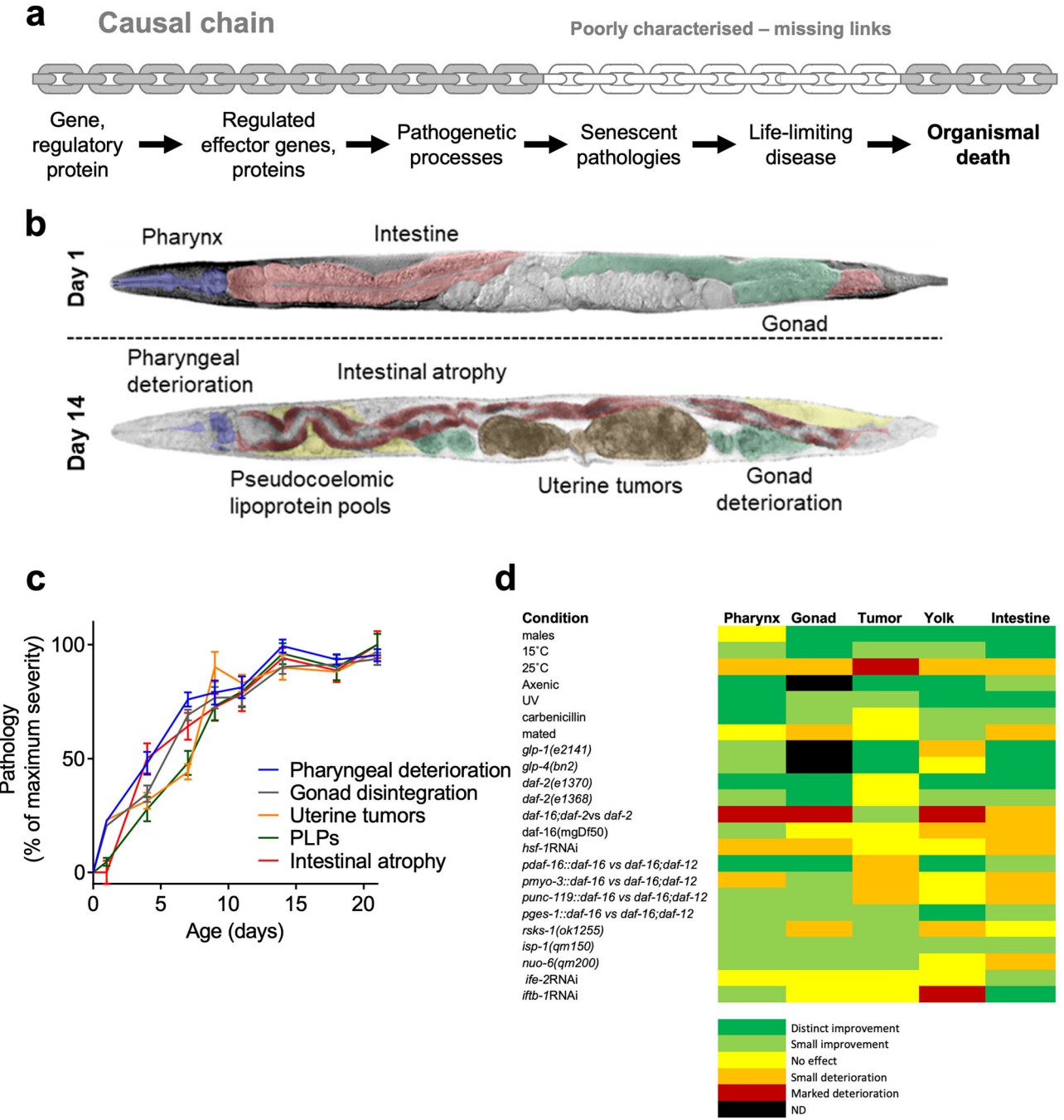

**Fig. 1 | Age-related pathogenesis is a missing link in the chain of events between gene action and lifespan. a** The causal chain of events between gene action, aging and lifespan, and the gaps in our understanding thereof. This knowledge gap currently limits what we can usefully learn from studies of aging in this animal model. **b** Pathological changes from day 1–14 visibly affect the pharynx, intestine, gonad, and uterus (uterine tumors) and cause ectopic deposition of lipid/yolk in the body cavity (pseudocoelomic lipoprotein pools, PLPs). DIC images of worms with artificial coloring of organs/pathological changes. Top: Young, healthy animal with intact tissues and no pathology. Bottom: Old animal with multiple pathologies.

**c** A surge in pathogenesis in early-mid adulthood. Development of age-related pathologies in *C. elegans* hermaphrodites starts in early adulthood, largely attaining their full extent by ~day 10. Error bars show S.E.M. **d** Heatmap of pathology development in experimental conditions used in the study, relative to control. Distinct improvement: pathology score reduced by >50%. Small improvement: pathology reduced by <50%. Small deterioration: pathology increased by <50%. Distinct deterioration: pathology increased by >50%. ND: Not determined. Control: standard culture conditions, wild type, 20 °C unless otherwise indicated.

generated for this study. For a listing of all data and sources, see Supplementary Data 1, and Supplementary Table 2. The data comprised pathology severity measurements and lifespan estimates, across 47 different genetic or environmental conditions. Also included was data from several species in the *Caenorhabditis* and *Pristionchus* genera; among these, hermaphrodites show higher levels of senescent pathology and are shorter lived than females of gonochoristic sibling species[7].

For data gathering, Nomarski microscopy was used to track the development of naturally occurring senescent pathologies. These can be readily assayed in vivo thanks to the optically transparent *C. elegans* cuticle. This approach builds upon earlier work[9,10,12,15–20] characterizing in particular, five prominent age-related pathologies that occur in aging wild-type *C. elegans*. These are deterioration of the pharynx, atrophy of the intestine, accumulation of yolk-rich pseudocoelomic lipoprotein pools (PLPs),

atrophy and fragmentation of the distal gonad, and development of teratoma-like uterine tumors (Fig. 1b, c, Supplementary Fig. 1a, b, Supplementary Table 1)[7,12,15,20,21].

Treatments for which new data was gathered included the following (Fig. 1d). Culture on axenic medium plates, a putative dietary restriction treatment (Supplementary Fig. 2); diverse manipulations of insulin/IGF-1 signaling (IIS), affecting the *daf-2* insulin/IGF-1 receptor and the *daf-16* FOXO transcription factor (Supplementary Figs. 3–5), and manipulations of *hsf-1* signalling (Supplementary Fig. 4f–j), protein synthesis, mitochondrial function (Supplementary Fig. 6) and the mTOR pathway (Supplementary Fig. 7). Pathology measurements were performed by sampling 10 animals from the population at multiple ages. Also included was a dataset from a cohort of wild-type animals, where the individuals were tracked throughout life, and pathology and lifespan were measured. This allowed a test of ML models in the absence of possible bias introduced by population sampling.

As expected, interventions varied in the severity of their effects on age-related pathology. Notably, not only did the degree of suppression differ between pathologies, but the relative degree of suppression also varied between treatments (Fig. 1d, Supplementary Figs. 3–7). Greater impacts were most often seen on the intestine and the pharynx, with weaker effects on uterine tumors. The presence of such variation provides a good basis for the application of ML to investigate how age-related pathologies affect lifespan in *C. elegans*. For further discussion of observed effects on pathology, see Supplemental Discussion.

## Differential correlations between pathologies and metrics of declining health

Senescence involves diverse changes that are pathological insofar as they disrupt biological function and, in some cases, contribute to late-life mortality. The age-related changes in anatomy documented here are clearly degenerative (organ atrophy, hypertrophy, and structural deterioration), which strongly suggests that they are pathological in nature[15,22]. Other possible criteria for identifying them as pathological are that they contribute to age-related decline in health and to late-life mortality. We first probed the former possibility.

As *C. elegans* hermaphrodites grow older, they exhibit various behavioral impairments affecting, among other things, defecation, locomotion, and pharyngeal pumping. Age changes in these three health-related parameters were measured, and the patterns of decline seen were comparable to those previously described[23–25]. Fig. 2a–c displays comparisons of age changes in three pairs of health metrics and pathologies in wild-type animals under standard culture conditions, while Fig. 2d shows a correlation matrix between health and pathology metrics.

Consider first the decline in defecation rate. This proved to be correlated most strongly with intestinal atrophy, as well as with pharyngeal deterioration and gonad atrophy (Fig. 2a, d). Here a plausible possibility is that intestinal atrophy impairs intestinal function, including defecation. Decline in locomotion proved to be most strongly correlated with uterine tumor development (Fig. 2b, d). Tumors frequently attain a great size, such that they fill much of the body cavity in the mid-body region; one possibility is that this impedes body bends, either by increasing mid-body rigidity or due to the tumor pressing against body wall muscles. These two examples are at least consistent with the view that intestinal atrophy and uterine tumor formation are pathological processes.

By contrast, pharyngeal pumping declined some time later than the appearance of pharyngeal pathology (Fig. 2c). This could imply that age changes in neuronal control of pumping rather than organ degeneration underlie the age-related decline in feeding rate. The co-occurrence of the mid-life anatomical changes and decline in movement and defecation provides further support for the designation of such changes as pathological.

## Negative correlation between age-related pathology development rate and lifespan

Next, we investigated the relationship between age-related pathology and lifespan. To begin with, we conducted an overall analysis of the correlation between lifespan and the overall average severity of all five pathologies using our datasets (Supplementary Table 2). As a metric of pathology severity, we measured the rate of pathology progression. To obtain the rate of pathology progression, all raw data were first transformed into Z-scores (which describe a value's relationship to the mean of a group of values). This was done to allow pathologies to be compared to one another; standard scoring systems that are used in the literature (and thus employed in this work) vary from one another, and so without this correction are not directly comparable. It should be stressed that such comparisons are only an approximation. Given the complexity of biological systems, it is currently not feasible to determine the full systemic effects of a specific pathology at a given level.

Following Z-score transformation, as one might expect, a negative correlation between mean pathology development rate and lifespan was observed, and this was consistent across all treatments and observed both in *C. elegans* and in the other nematode species used in this work ($R^2 = 0.5$; Fig. 2e). Z-score transformation is appropriate in this case, given the synchronous nature of pathology development in *C. elegans* (Fig. 1c). This is consistent with the possibility that age-related pathology contributes to late-life mortality. Given that this is only an approximation, and considering the complexity of biological interactions alongside the challenge of determining whether all pathologies contribute equally to survival, or alternatively whether a subset is the predominant determinant of lifespan, machine learning (ML) was applied to explore this further.

## Use of machine learning to identify potential life-limiting pathologies

In line with our predictions, results in this study underscore how different treatments that extend lifespan differentially suppress age-related pathologies (Fig. 1d, Supplementary Fig. 3-7). This raises the possibility that it is a combination of pathologies, in an independent or dependent manner, that affects lifespan. The latter would mean that the effect of one pathology on lifespan may be influenced by one or more other pathologies; such interactions could, in principle, follow any number of relationships, ranging from linear to non-linear.

To investigate the relationship between pathology severity and lifespan, an ML-based approach was employed. To obtain a dataset to interrogate that was of a sufficiently large size for ML analysis, we combined data generated in this study with various data from previous studies, from our lab and others, as described above (Supplementary Table 2). The dataset included a total of 434 observations across 23 conditions and 8 different species, up to day 11 of adulthood, by which age pathologies have usually reached maximum severity[15].

First, we randomly split the data into two groups: (i) one group of 80% to build the model; and (ii) the remaining non-overlapping 20% of data to be used to validate the model. We trained different standard models (linear regression, ElasticNet regression, Support Vector Machine [SVM], random forest [RF], and multilayer perceptron [MLP]) using the 80% training set (raw pathology scores and corresponding lifespan data). We then evaluated the models using the unseen 20% testing set to identify which best predicts lifespan based on pathology at all ages assayed. The random forest (RF) model outperformed the others in predicting lifespan ($R^2 = 0.57$; mean average error [mae] of predicted life: 4.0 days) (Supplementary Fig. 8a).

Next, we needed to account for the fact that the pathologies progress with time and that pathology values at different time points are not independent of one another. In other words, we needed a means to account for the incremental progression of pathology from one time point to the next. To this end, differences in scores through time were fed into the different models (i.e., pathology scores at each time point had the previous time point scores subtracted), with training and evaluation of the models as described above. Again, the RF model outperformed the others, yielding an $R^2$ of 0.79 and mae of 2.77 days for scores measured on day 11 (Fig. 3a). In other words, 79% of the variation in nematode lifespan between different contexts can be predicted using the five age-related pathologies measured here. This was despite the high diversity in lifespan of the populations studied, which

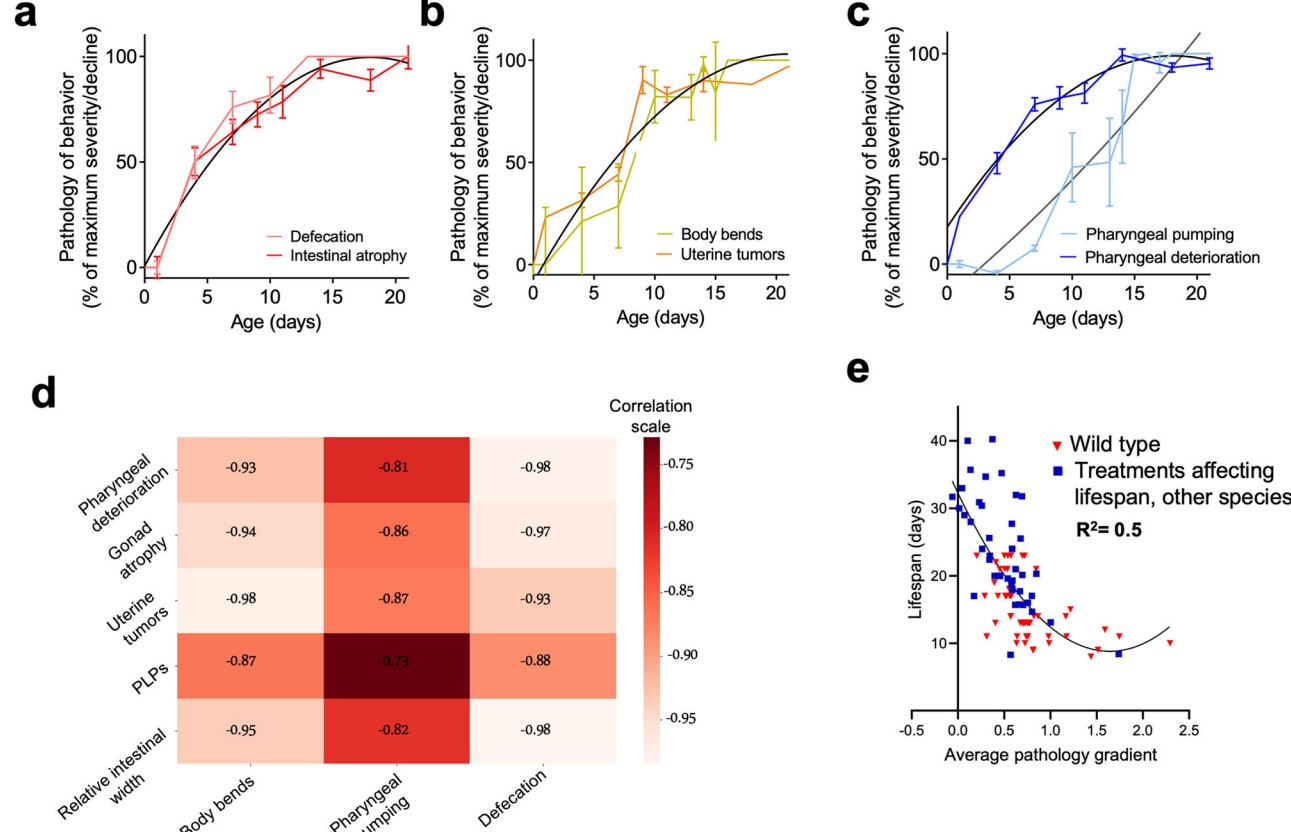

**Fig. 2 | Variation in age-related pathology with health metrics.** Health metric declines in early-mid adulthood. Selected comparisons of fits (extra sum-of-squares F test). **a** Close correspondence of decline in defecation with intestinal atrophy, and (**b**) of decline in movement (body bends) with uterine tumor development. **c** Loss of pharyngeal pumping occurs later than the main period of pharyngeal pathology development. **a**–**c** 2 trials ($n = 10$/trial), error bars show S.E.M. **d** Correlation matrix of the different pathologies measured and decline in health metric (analysis of raw data). Pearson method. **e** Broad correlation between pathology and lifespan. Blue symbols, population data; red triangles, individual data. Pathology progression through time is converted into a gradient and then transformed into Z-scores (which describe a value's relationship to the mean of a group of values) for standardization.

This allows comparison between different pathologies by normalizing levels of a given pathology to the average level of that pathology in the group (represented by a Z-score of 0, with a Z-score of +2 representing the maximum pathology severity in the group, and −2 the healthiest animals). Intestine scores were normalized to the respective mean intestine score on day 1 prior to being converted to a gradient, such that the normalized intestinal pathology scores represent the change in percentage of intestinal volume. This takes into account differences in terms of the ratio of intestinal width to whole body width between different treatments and species. The average of all 5 pathology Z-scores is shown (x-axis) plotted against the real observed lifespan (y-axis).

ranged from 7.5 to 40 days (mean: 18.3 days, lowest quartile: 13 days; Fig. 3b).

Next, we asked which pathologies contribute the most to lifespan, according to our model. In terms of model feature importance, the pharynx and intestinal pathology scores correlate the most strongly with observed lifespan (Fig. 3c). Furthermore, the greatest determinant of model prediction (based on Gini index or Mean Decrease in Impurity [MDI]) is pharyngeal pathology, by a wide margin (Fig. 3d, Supplementary Fig. 8c). This is consistent with previous findings pointing to links between lifespan and both pharyngeal and intestinal status[7,15,26,27].

We also looked at differences between longer- and shorter-lived single wild-type worms, as well as longer- and shorter-lived treatment populations and species used in the model. Here, longer lived was defined as >18 days and shorter lived <18 days. This value was selected as it is the approximate mean *C. elegans* lifespan at 20 °C as well as the mean of all single wild-type worms, treatments (culture under standard vs non-standard conditions, e.g. axenic medium) and species used in this study (Fig. 3b). This analysis revealed that pharyngeal pathology is more predictive of lifespan in shorter-lived animals than in longer-lived ones (Fig. 3d). One possibility is that this reflects increased early deaths linked to pharyngeal infection[26]. By contrast, intestinal atrophy and PLPs are similarly predictive of life in shorter- and longer-lived animals. Notably, uterine tumors were more predictive of

lifespan with treatments that extend life. This suggests that in animals subjected to life-extending treatments, the contribution of uterine tumors to late-life mortality increases.

Finally, to assess the predictive potential of this model for future datasets, we collected lifespan and pathology data from two new conditions not previously used in the model (creation or validation): wild-type animals cultivated with *Stenotrophomonas* MYb57 and *Achromobacter* MYb9, two bacterial isolates identified in wild *C. elegans* and shown to colonize the *C. elegans* gut[28,29]. Notably, predicted vs actual lifespan showed no significant difference across treatments, and predicted lifespans were within 1 day of empirically observed lifespans (Supplementary Fig. 9), supporting the predictive power of the ML model.

**Sexual dimorphism in development of age-related pathologies**

While conducting the above analysis, other observations relating to senescent pathology were made. In *C. elegans*, most aging research is directed at hermaphrodites, while relatively little is known about aging in males. Some sex differences have been previously noted, as follows. Males cultivated in isolation live longer than hermaphrodites[30]. Aging males do not accumulate yolk pools in the body cavity[15], consistent with the absence of vitellogenin (yolk protein) synthesis in males[31]. Moreover, males do not exhibit distal gonad atrophy[32] or marked intestinal atrophy[15].

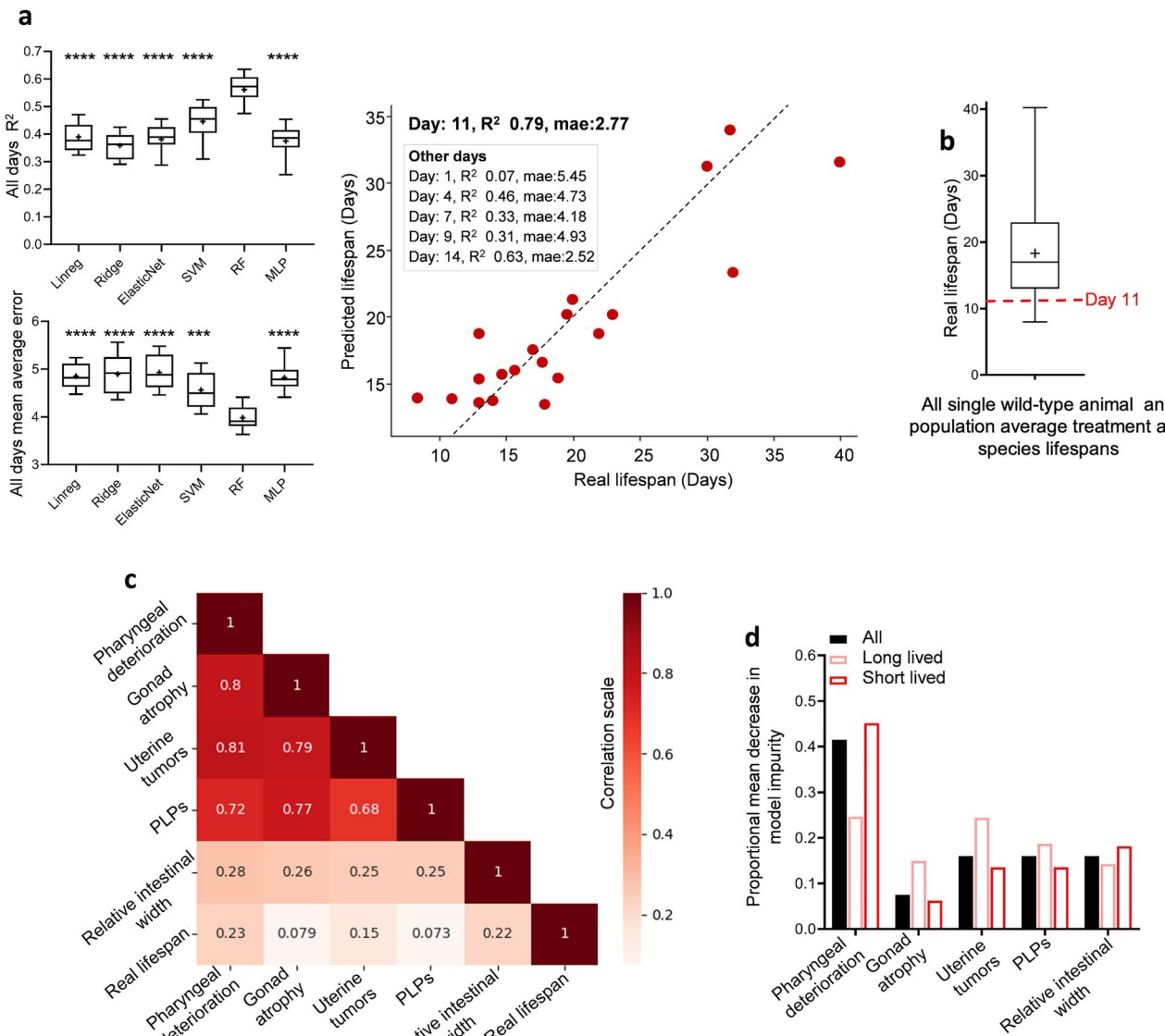

**Fig. 3 | Machine learning can predict lifespan from age-related pathology.**
**a** Selection of a random forest (RF) machine learning (ML) algorithm as the best predictor of lifespan from pathology, and use of the RF model to predict lifespan with $R^2 = 0.79$. Left: The RF model was identified as the best predictor of lifespan. $R^2$ values and mean absolute error (in days) are shown for different ML models, including linear regression (Linreg), Ridge regression (Ridge), ElasticNet regression, Support Vector Machine (SVM), RF, and multilayer perceptron (MLP). All pathology measurements conducted up to day 14 of *C. elegans* lifespan were used. Day 14 was chosen because it is the time point at which maximum pathology severity was observed across all animals (see Fig. 1c). Right: Lifespan prediction using the RF model achieved $R^2 = 0.79$. This analysis was repeated using the RF model, but instead of using all data up to day 14, data from up to various earlier time points (i.e., up to day 1, 4, 7, 9, 11, and 14 of adulthood) were used separately. The $R^2$ for each is shown in the inset. Day 11 outperformed all other time points, which was predicted, as it represents the time point where pathology severity is generally highest before pla-teauing at its maximum around day 14 (see Fig. 1c). The scatter dot plot shows data only up to day 11 of *C. elegans* lifespan. Each dot represents a held-out test sample and shows observed lifespan versus lifespan predicted by the RF model using pathology data. Eighty percent of the animal samples were used to train the model, and the remaining 20% were used to evaluate it. Each sample corresponds either to a single animal (n = 55 wild-type individuals) or to the mean value of a population subjected to a specific treatment or genotype (n = 45 population means) observed at different intervals (see Methods). For comparison of different models using only data up to day 11 of adulthood, see Supplementary Fig. 8b. For coefficients of the linear regression model, see Supplementary Fig. 8b. *p*-values compare the RF model to other models. **b** All wild-type single animal lifespans, and average population lifespans of various treatments and species used to build the model plotted as a box plot. The average lifespan of all single wild-type worms, treatments, and species is 18.3 days. Note that some treatments extend lifespan while others shorten lifespan. Day 11 (shown in red) is the time point up to which pathology progression was scored and used to predict lifespan in (**a**). **c** Correlation matrix of the different pathologies measured (raw data) and the real observed lifespan. Pearson method. **d** Feature importance: RF model Mean Decrease in Impurity; long-lived refers to worms having survived for more than 18 days. *t*-test: long-lived vs short-lived animals. Note that the Pearson correlations shown in (**c**) quantify only linear pairwise associations, whereas the Random-Forest feature-importance scores in (**d**) (mean decrease in impurity) reflect non-linear effects and higher-order interactions; consequently, a variable with a low linear correlation can still rank as highly important in the multivariate RF model. ***$p < 0.0001$, ****$p < 0.00001$. For raw pathology and lifespan data used see Supplementary Datasets 1 and 2.

To give a fuller picture of sex differences in age-related pathology, we directly compared all five pathologies in the two sexes (unmated) (Fig. 4a–e). This confirmed the absence of yolk pools, intestinal atrophy, and gonad atrophy in males. Moreover, we detected no form of germline tumor equivalent to the teratoma-like uterine tumors seen in aged hermaphrodites[20,33]. By contrast, similar levels of pharyngeal pathology were seen in the two sexes (Fig. 4a–e). Thus, major pathologies develop rapidly in organs linked to reproduction (i.e., the gonad and intestine) in

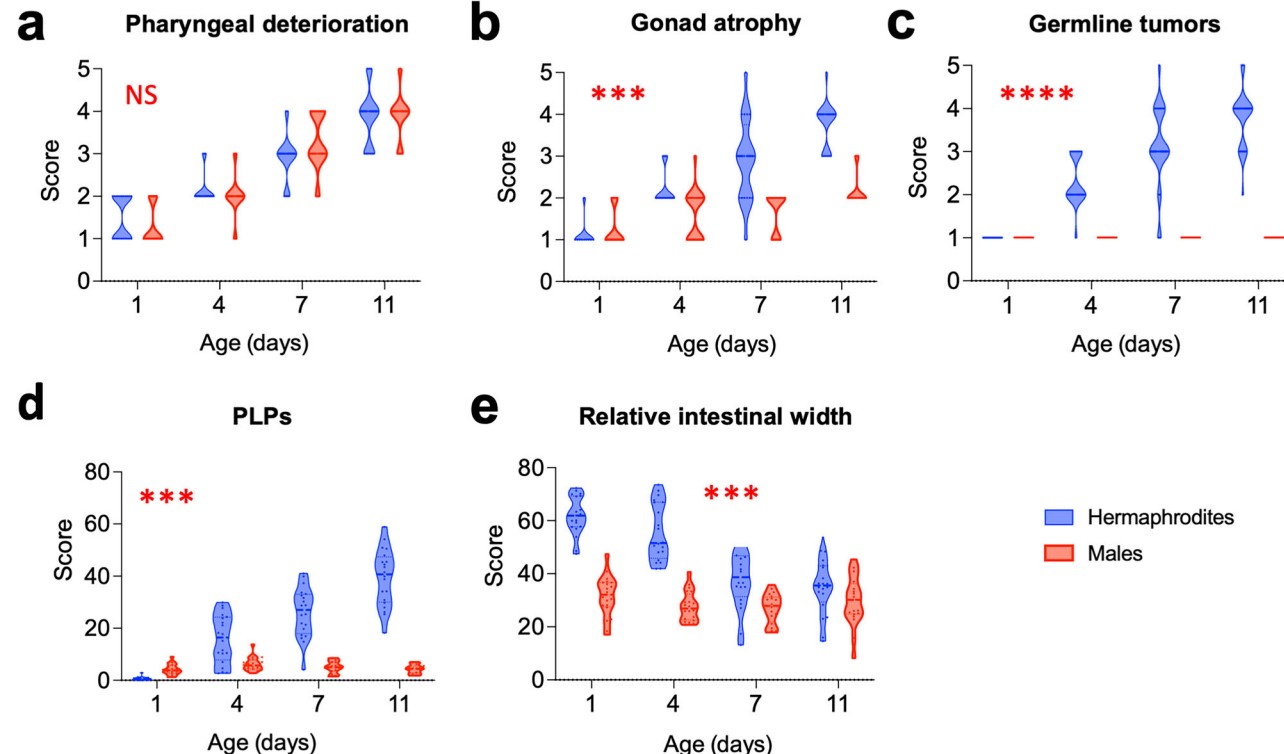

**Fig. 4 | Sex difference in age-related pathology.** Most of the age-related pathologies seen in hermaphrodites are largely absent from males (two-way ANOVA, Bonferroni correction); stars show statistically significant differences in pathology progression (ANCOVA, Tukey correction). **a** Pharyngeal deterioration is similar in both sexes. **b** Gonad atrophy is absent from males. **c** Germline tumors (teratoma-like uterine tumors in hermaphrodites) and (**d**) PLPs (yolk pools) are absent from males. **e** Gut atrophy is largely absent from males. 2 trials ($n = 10$/trial). ***$p < 0.0001$; ****$p < 0.00001$.

hermaphrodites but not males. This could imply that trade-offs linked to reproduction lead to pathology in hermaphrodites but not males. These results demonstrate marked sexual dimorphism in age-related pathophysiology in *C. elegans* and, therefore, potentially, in the causes of late-life mortality, and hint at a greater role of pharyngeal pathology in late-life mortality in males than in hermaphrodites.

## Discussion

Interventions that alter lifespan provide a possible means to understand mechanisms of aging. Achieving this requires an account of how such interventions affect senescent pathology and how senescent pathology affects mortality. Towards such an account, we have described how effects of interventions that alter lifespan have differential effects on age-related pathology in *C. elegans*. Capitalizing on this, we have investigated the relationship between pathology and lifespan using a ML approach. The results suggest that pharyngeal and intestinal pathology are significant determinants of late-life mortality. Moreover, in life-shortening contexts, the importance of pharyngeal pathology increases (and potentially in males too), while that of uterine tumors increases in life-extending contexts.

### Use of ML analysis: insights and limitations

The use of several ML models to predict lifespan from pathology was tested, and one, the random forest (RF) model, consistently outperformed the others. Our analysis found that non-linear ML models consistently outperformed the linear ones when predicting lifespan. In particular, and notably, RF regression models based on changes across time points (rather than absolute pathology scores alone) yielded the best performance. In line with our predictions, this suggests a contingent nature of the relationship between pathology and survival, that the informativeness of pathology scores varies with age, and that scores for different pathologies can be interdependent. Rather than linear or additive relationships, the results support the prediction that given pathologies only limit lifespan at specific

severities and in specific combinations with one another. We suggest, therefore, that it may be for this reason that the creation of decision trees by the RF model has the greatest predictive power. (Decision trees are where the model learns how best to split the dataset into smaller and smaller informative subsets to predict the target value).

The models are able to predict 79% of the variation in lifespan, but what about the remaining 21%? Here it should be noted that our dataset included data compiled from other studies where minor differences in culture conditions could create noise. For example, the mean lifespan of *C. elegans* under standard conditions at 20 °C typically ranges from 17 to 21 days. Moreover, pathology is typically scored only every 3–5 days. Furthermore, this study was limited to five pathologies, which are easily assessed in vivo without the use of reporters or staining methods. Other, less visible pathologies are likely to contribute to mortality. Not included here are, for example, muscle atrophy, cuticular hypertrophy[9], and age-related changes in the nervous system such as neurite outgrowths, synapse deterioration[34,35], and PVD neuron degeneration[36]. Considering these sources of noise, an $R^2$ of 0.79 is a notably high degree of correlation, comparable to what has been found when using aging biomarkers combined with ML to predict age in humans ($R^2$ between 0.80 and 0.83)[37]. In the future, the approach taken here could be developed further by including more pathologies to obtain improved predictions. It should be emphasized, however, that feature importance in ML models does not prove causation, but rather is an indicator of predictive ability; to demonstrate causation will require future direct tests of effects of pathology on late-life mortality. More broadly, the application of ML approaches to understand *C. elegans* aging is a topic of growing opportunity[38].

### Is lifespan a function of overall aging or specific life-limiting pathologies in *C elegans*?

The correlation between intestinal pathology and lifespan supports a central role for the intestine in nematode aging. This could be because the gut

regulates aging in the whole organism by activating signalling pathways in other tissues. Notably, rescue of DAF-16 activity in the gut partially restores *daf-2* mutant longevity in *daf-16; daf-2* mutants. There is evidence that this involves paracrine action via both DAF-16-dependent and -independent responses, involving alteration of gene expression[39–41] and of various processes in other tissues[42–44], reviewed by Ref [45]. Thus, the intestine may regulate aging in the whole organism via paracrine (and also autocrine) signaling.

A second possibility is that intestinal pathology (including atrophy) directly increases mortality. Limitation of life by organ-specific pathology is a plausible explanation for the correlation of pharyngeal deterioration with mortality, given that suppression of early death from pharyngeal *E. coli* infection increases lifespan[26,27,46]. If intestine-limited rescue of *daf-16* in *daf-16; daf-2* double mutants extends lifespan mainly by regulating whole-animal aging, one would expect to see amelioration of multiple pathologies. By contrast, if it acts by preventing life-limiting pathology in the gut, one would expect little effect on other pathologies. In fact, the latter was the case (Supplementary Fig. 5), suggesting that intestinal pathology contributes to late-life mortality. Moreover, intestinal permeability increases late-life mortality in *C. elegans*[47], and it is likely that intestinal atrophy results in increased permeability. Thus, it is likely that life-extension from intestinal rescue of *daf-16* (Ref [39]) and of other factors[15,48] reflects, at least in part, rescue of life-limiting intestinal pathology.

### Context-dependent effects of pathology on late-life mortality?
Differences in pathology-lifespan correlations were observed between short-lived and long-lived populations. In the former, pharyngeal pathology correlates best with lifespan, while in the latter, the same is true of uterine tumors. In line with our predictions, this suggests that effects of pathology on late-life mortality can exhibit context-dependency, i.e., show effects in some conditions but not others. Notably, it has been observed that longer-lived *C. elegans* spend a larger proportion of life in a senescent, immobile state, which may in part reflect resistance to colonization of the pharynx by *E. coli*, allowing survival to later, frailer ages[26,49,50]. Thus, pharyngeal deterioration may disproportionately increase vulnerability of shorter-lived animals to life-shortening bacterial infection. In contrast, uterine tumors may contribute to mortality more in animals that have escaped infection and are long-lived[17,51]. This could imply that amelioration of pathologies that limit life leads to their replacement by new causes of late-life mortality, here bacterial infection and uterine tumor development, respectively[52]. In the latter case, a possibility is that life-shortening secreted uterine proteins play a role[53].

### Sexual dimorphism in pathology development and aging
The aging process exhibits sexual dimorphism in a number of ways, ranging from sex differences in evolutionary determinants of aging[51], in effects of interventions that increase lifespan, and in patterns of aging and late-life disease in men and women[52,54]. Our data shows major sex differences in age-related pathologies in *C. elegans*: the severe age-related pathologies seen in hermaphrodites are largely absent in males, with the exception of pharyngeal deterioration (Fig. 4a–e). This demonstrates fundamental differences in the aging process between hermaphrodites and males.

Coincident with their reduced pathology, the lifespan of wild-type males is typically about 20% longer than that of hermaphrodites, at least when life-shortening male-male interactions are prevented[30,55–57]. Possibly this reflects the reduced levels of age-related pathogenesis in males, apart from that affecting the pharynx. The latter suggests that pharyngeal senescence may play a greater role in late-life mortality in males.

Recent evidence suggests that the severe pathologies in hermaphrodites reflect the occurrence of semelparous reproductive death[7,58,59]. The paucity of this pathology in males suggests that reproductive death in *C. elegans* is hermaphrodite specific. Consistent with this, prevention of reproductive death in semelparous species can result in large increases in lifespan[7], and in *C. elegans* germline removal (which greatly reduces age-related pathology in hermaphrodites) increases lifespan in hermaphrodites[60,61] but not males[62].

## Methods
### Culture methods and strains
*C. elegans* were maintained at 20 °C on Nematode Growth Medium (NGM) plates seeded with *Escherichia coli* OP50, unless otherwise stated. 5-fluoro-2-deoxyuridine (FUDR), sometimes used to block progeny production, was not used in this study. *C. elegans* strains used in this study included: N2 hermaphrodite stock, N2M male stock *fln-2(ot611) X* (ref.[47]), CB3844 *fem-3(e2006) IV*; CF1724 *daf-16(mu86) I; daf-2(e1370) III; muIs105 [daf-16p::GFP::daf-16 + rol-6(su1006)]*, CF2005 *daf-16(mu86) I; daf-2(e1370) III; muIs120 [ges-1p::GFP::daf-16 + rol-6(su1006)]*, CF2093 *daf-16(mu86) I; daf-2(e1370) III; muIs131 [unc-119p::GFP::daf-16 + rol-6(su1006)]*, CF2102 *daf-16(mu86) I; daf-2(e1370) III; muIs126 [myo-3p::GFP::daf-16 + rol-6(su1006)]*, DR466 *him-5(e1490) V*, DR1563 *daf-2(e1370) III*, DR1572 *daf-2(e1368) III*, GA14 *fog-2(q71) V*, GA114 *daf-16(mgDf50) I; daf-2(e1370) III*, GR1307 *daf-16(mgDf50) I*, MQ887 *isp-1(qm15) IV*, MQ1333 *nuo-6(qm200) I* and RB1206 *rsks-1(ok1255) III*.

Pathology and survival data used in ML analysis were drawn from both this study and a number of earlier studies. For the latter, strains and sources are listed in Supplementary Table 2.

### Preparation of aging cohorts for pathology measurements
For each condition, approximately 200 animals were aged on 4 different plates, starting from the pre-adult L4 stage. Animals were transferred to fresh plates daily during reproduction (to avoid overgrowth with offspring) and at approximately weekly intervals thereafter. In trials using RNA interference (RNAi), HT115 RNAi-producing (clones *ife-2, iftb-1, hsf-1*) and control bacteria (plasmid L4440), were cultivated as previously described in ref. [63], washed and seeded onto NGM plates. RNAi treatment was started at the L4 stage. For trials involving males, animals were maintained at low population density ( ~ 5 per plate) to reduce detrimental effects of male-male interactions[30] and transferred to fresh plates at the same time as hermaphrodites.

### Pathology measurements
Animals were imaged on days 1, 4, 7, 11, 14, and 18. At each time point, 10-15 animals were mounted onto 2% agar pads and anesthetized with 0.2% levamisole. DIC images were acquired with an Orca-R2 digital camera (Hamamatsu) and either a Leica DMRXA2 microscope or a Zeiss Axioskop 2 plus microscope, driven by Volocity 6.3 software (Improvision, Perkin-Elmer). Images of pathology were analysed semi-quantitatively[10,15,21] (Supplementary Fig. 1).

For pharyngeal, gonadal, and uterine pathologies, images were randomized, examined by trained scorers, assigned severity scores of 1–5, and mean values calculated. Here 1 = youthful, healthy appearance; 2 = subtle signs of deterioration; 3 = clearly discernible, mild pathology; 4 = well-developed pathology; and 5 = tissue so deteriorated as to be barely recognizable (e.g., gonad completely disintegrated), or reaching a maximal level (e.g., large uterine tumor filling the entire body width). Intestinal atrophy was quantified by measuring the intestinal width at a point posterior to the uterine tumors, subtracting the width of the intestinal lumen, and dividing by the body width. Yolk accumulation was measured by dividing the area of pseudocoelomic yolk pools by the cross-sectional area of the body visible in the field of view as captured at 630x magnification. Animals were discarded after imaging.

### Single worm pathology and lifespan measurements
Animals were cultivated individually starting from L4 stage. Single worm pathology measurements were performed on days 1, 4, 7, 11, 14, and 18. Animals were mounted onto 2% agar pads on a microscope slide, and the slides placed on ice for 5 min to prevent animals from moving during imaging. Animals were then immediately imaged and recovered from the slide by carefully removing the coverslip and transferring from the agar pad using an eyelash pick. Lifespan was also measured in each animal.

https://doi.org/10.1038/s42003-025-09012-9 **Article**

## Axenic culture

For experiments using axenic medium plates, with no *E. coli*, plates were prepared as previously described in ref. 64. Briefly, gravid hermaphrodites were bleached and eggs were plated on NGM plates bearing UV-irradiated *E. coli*, to allow normal growth to the L4 stage. L4 larvae were then transferred to and maintained on axenic medium plates.

## Statistics and reproducibility

Pathology development was analysed using two-way ANOVA to compare different time points. Differences in pathology progression were compared using an ANCOVA, with Z-scores for standardization of different pathologies, as described[7]. Bonferroni's or Tukey's tests were used to correct for multiple comparisons.

## Machine learning model generation

Machine Learning (ML) regression models were trained to predict the lifespan of nematodes (individual wild-type animals under standard conditions, or populations with altered genotypes/culture conditions) based on pathology development features. The severity of pathologies was scored on days 1, 4, 7, 11, and 14 (for some conditions in which lifespan is reduced, scoring on day 14 was not possible). For individuals, wild-type worms were followed throughout life, and day of death was scored, as well as their pathology levels. For populations, pathology was scored in samples of 10 individuals at each time point, and the mean lifespan of the population was used.

More precisely, ML models were trained to learn a mapping $f : X \rightarrow y$. Here $X = [x_1 \ldots x_{N_F}]$ represents the observed pathology development features (number of features: $N_F = 5$): pharyngeal deterioration, gonad atrophy, uterine tumors, PLPs, and intestinal width, observed at each time point. $y$ represents the lifespan of a particular worm individual or population. In total, 438 observations were used to train and evaluate the ML models. A small percentage ( < 10%) of pathology development values were missing due to measurement and recording errors (e.g., where captured images were too unclear to score pathology). These missing values were replaced with the mean values of the pathology feature observed on a specific day. For example, the missing pharyngeal deterioration value of a worm observed at day 7 was set to the average of all pharyngeal deteriorations observed at day 7. The target value $y$ for the ML model predictions was the observed lifespan of the worm minus the day at which the pathology features were observed.

Three linear ML models (Linear Regression, Ridge Regression, and ElasticNet regression) and three non-linear models (Support Vector Machine ['rbf' kernel], Random Forest [500 estimators], and a Multilayer Perceptron with one hidden layer ([5 hidden nodes, Tanh activation function]) were evaluated. The models were implemented in Python using the scikit-learn toolbox[65]. The data comprises 438 samples × 5 pathology features observed corresponding to either a single animal ($n = 55$ wild-type individuals) or the mean value of a population subjected to a treatment/genotype ($n = 45$ population means) observed at different intervals (days 1,4,7,9,11,14). The data was randomly split (80:20) to generate an 80% training set and 20% testing set. Each model was trained and evaluated with 20 random train-test (80:20) splits.

## Supplementary information

Supplementary data files include: Supplementary Information. Supplementary Dataset 1: Raw pathology and lifespan data original to this study. Supplementary Dataset 2: Raw data used to build the ML model.

## Reporting summary

Further information on research design is available in the Nature Portfolio Reporting Summary linked to this article.

## Data availability

Raw data used to generate the ML model is available in the Supplementary datasets. All other data are available from the corresponding authors upon request.

## Code availability

Code used to generate ML model available on: https://github.com/UCL/celegans-lifespan.

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

## Acknowledgements

This work was supported by a Wellcome Trust Investigator Award (215574/Z/19/Z) to D.G. and a BBSRC Research Grant (BB/V011243/1) to M.E. We thank Jennifer Tullet (University of Kent) and members of the Gems lab for useful discussion and/or comments on the manuscript. Some strains were provided by the Caenorhabditis Genetics Center, which is funded by NIH Office of Research Infrastructure Programs (P40 OD010440).

## Author contributions

M.E., D.G. and C.C.K. contributed to the wet lab project design. C.C.K. developed the theory behind mathematical and ML analysis and performed data interpretation. C.A., M.E., A.F.G., C.C.K., H.W., S.S. and A.Z. performed wet lab data collection. P.M.S. and M.C. performed ML data tabulation, code creation and analysis. D.G. initially drafted the manuscript, and M.E., D.G. and C.C.K. edited the manuscript.

## Competing interests

C.C.K. is CEO of the biotech company LinkGevity. The other authors declare no competing interests.
