## [Transparent Peer Review file · Communications Biology]

Machine learning predicts lifespan and underlying causes of death in aging *C. elegans*

Corresponding Author: Dr Marina Ezcurra

Version 0:

Reviewer comments:

Reviewer #1

(Remarks to the Author)

In this manuscript #COMMSBIO-25-1850-T by Carina C. Kern et al. entitled 'Machine learning predicts lifespan and underlying causes of death in aging *C. elegans*', the authors describe the development and use of a new machine learning model to question the theoretical prediction that age-related pathologies are context-dependent. The authors propose to study how interventions altering lifespan affect pathology, and how pathology affects lifespan. To do this, the authors selected variables for evaluating the senescence process in the *C. elegans* model from observations taken from the scientific literature, as well as a large volume of observations presented in the manuscript.

By using direct observations with Nomarski microscopy, the authors selected the following as markers of senescence: pharynx, atrophy of the intestine, accumulation of yolk-rich pseudocoelomic lipoprotein pools, atrophy and fragmentation of the distal gonad, and development of teratoma-like uterine tumours. By establishing scores from 1 to 5 for each of the age-related pathology markers, the authors were able to train the machine learning model with up to 434 observations of wild nematodes as well as different mutant lines in combination with functional integrity markers of muscle (body bends, pharyngeal pumping) and intestine (atrophy and defecation). The observations thus describe the impact of mutants in the *Daf-2/Daf-16* aging regulatory pathway, *Hsf-1*, *mTOR* and the mitochondrial respiratory chain on the degree of severity of age-related pathology markers. Tissue rescue experiments specific to *DAF-16* function allow the authors to discuss the hierarchy of tumours observed in the control of senescence. The impact of axenic feeding on the longevity of *C. elegans* is very well documented and has been evaluated in this work, providing excellent complementary results that provide a broader picture of explanations for the effects observed on the age-related pathology markers of the digestive apparatus. A clear negative correlation between age-related pathology markers occurrence and lifespan is described. Then, the authors ask if the observed age-related pathologies can appear independently of each other or in a dependent manner and propose, in fine, that pharyngeal deterioration plays a more important role than the other age-related pathology markers. A comparative analysis of age-related pathology markers between the two sexes of *C. elegans* enabled the authors to clarify the different evolution of senescence and the impact of certain specific tissues in the process. Finally, using this new model based on five-related pathologies to predict lifespan with an accuracy of 79%, the authors propose a two-stage model of worm hermaphrodites.

In this manuscript, the authors propose an innovative approach in an attempt to better define the order of events and the main tissues responsible during the senescence process in the *C. elegans* model. This well-balanced manuscript represents the sum of high-quality experiments that have been carefully conducted. The results are very well discussed and benefit from a very critical approach that allows the reader to situate himself in the experimental strategy to answer the questions addressed.

The observation that this new tool can already predict senescence in this model up to a degree of 79% is highly original and, as proposed in the discussion section, points to the possibility of future improvements, such as taking into consideration other age-related pathology markers like sarcopenia, which can also be observed in this model. The statistical analyses, the number of participants and the experimental descriptions are described very precisely and are fully accessible to the experimenter.

I recommend the publication of this manuscript subject to the minor modifications as listed below:

In the Supplementary section of the manuscript

Supplementary fig.1a shows day 1 and day 14. To what age in days do the score 2, 3 and 4 images correspond? Please indicate this information on fig.1a.

In the references section of the manuscript

Names appearing in reference 17 in the references section of the manuscript must be corrected to appear with a capital letter at the beginning and in lower case thereafter.

The characters '<scp>' and '</scp>' in reference 38 should be deleted.

The title of the article in reference 51 should be corrected to lower case.

The names of the authors in capital letters in reference 54 must be corrected.

Reference 61 only includes the title of the article?

Names appearing in reference 63 in the references section of the manuscript must be corrected to appear in upper case at the beginning and in lower case thereafter.

Names appearing in reference 70 in the references section of the manuscript must be corrected to appear in upper case at the beginning and in lower case thereafter.

Reviewer #2

(Remarks to the Author)

This manuscript describes the development of a random forest model that predicts nematode worm life-span based on measurements of 5 age-related pathologies measured at Day 11. Based on the features that contributed most to predictions the authors suggest that diseases of the pharynx and intestines likely contribute to death in elderly *C. elegans*.

The manuscript is clearly and succinctly written, making it easy to review.

Overall, the methods used are methodologically and statistically sound, and the results are novel and interesting. That being said, I do have minor comments regarding terminologies, figure presentation and methods.

1. Although the text is nicely written, clear and concise, some of the figures could be improved. Particularly Fig2d needs resizing so the numbers and labels are visible. Y-axis labels are absent in some figures, and it is not always clear what the error bars represent (eg Fig1C).

2. The definition/public understanding of the word 'AI' has changed somewhat over recent years, and varies depending on who you ask. I personally, would not call these methods AI, and would prefer it be called only Machine Learning. I understand the temptation to refer to this as AI, since AI is a hot topic, but this paper is already great and does not need pumping-up with buzz words.

3. Please double check how you calculated your feature importance scores in Figure 3e, I am concerned a mistake has been made. In figure 3e you show the mean importance metrics for each feature over the 20 'model runs' which I assume means 20 train:test data splits? Your plots contain error bars showing showing the s.e.m over those 20 splits, however, each run is so similar that the error bar is not visible, suggesting the importance metrics are identical across all 20 runs. Generally, splitting the data will have some impact on feature importance. It seems unusual for each split to be identical. Please double check your code for how you calculated these importance scores. If your scores are identical in all 20 splits it could suggest a problem in your code. If your procedure is correct, please show these results in a way that shows the importance metric for each run separately, like a jitter plot. (i.e. show 20 small dots for feature to better visualize how the importance varies across the 20 data splits).

4. In your section 'Use of ML analysis: insights and limitations' please mention that feature importance in ML models does not prove causation, it just indicates predictive ability. To show causation, future analysis will need to include 'causal inference' methods.

Reviewer #3

(Remarks to the Author)

Several figure panels were so pixelated that they were essentially unreadable. Not sure whether this was due to a conversion issue or to the original files, but it made it challenging to follow arguments.

Kern et al. presented the use of machine learning to predict lifespan based on scores of several aging-associated pathologies in *C. elegans* with help from other species. Overall, it is a laudable effort, but I have substantial concerns about how the results are presented (text and figures). The paper was not easy to follow. Results were presented in a telegraphic fashion. In several places I was not readily able to understand which data were used in the analysis, what assumptions were made, etc. For a study that reports a complicated set of ML models, this is a serious problem that makes it virtually impossible to evaluate central claims. This interesting and useful study is almost inaccessible. Please rewrite it!

Major:

- Line 105/106. The statement ending with “largely self-evident, hence their description as pathological” struck me as underdeveloped. Do senescent pathologies cause death or are they byproducts of processes that cause death? Are they actually pathological? These distinctions are hard to make, but they are not trivial. It would not be reasonable to ask the authors to solve this problem, but a clear acknowledgement is necessary. The authors almost imply that they will present some evidence to back up their “self-evident” assertion, but in the page that follows (p.5) they do nothing of the sort. I don’t propose any specific experiments to substantiate the pathology claims, and the ability of the models to predict lifespan from pathology scores shows that they carry some relevant information, but I think it would serve the authors well to present a more seriously grounded argument here.
- The analysis described in line 127 and thereafter may be problematic. Although it may be valuable to compare progression of different pathologies, I am not convinced that the scheme proposed here would necessarily do so. First, for each trait an arbitrary scoring scheme was used. This is not to say that the chosen categories were illogical or wrong. Rather, no information was provided whether a step up from score 2 to score 3 was a “similar gain in pathology” for Traits A and B. Moreover, even if the scales were perfectly constructed, distributions of pathology scores among individuals could be considerably different for Traits C and D. This would limit the value of using Z scores. Neither of these two major problems was seriously addressed in the manuscript, rendering Figure 2e at least unsubstantiated and more likely uninterpretable. One of several reasons to be concerned is that the overall picture of multiple traits may be largely driven by one or two traits because their variability and the scoring scheme made them the dominant drivers of the overall score. The authors may be better served by omitting this line of argument altogether.
- Line 156. “randomly split the data into two groups: (i) one group of 80%”. REALLY unclear! What exactly do you mean by “the data” here? What does 80% refer to? A subset of all experiments? A subset of animals from each of experiments? Here and elsewhere, providing specific descriptions is essential.
- A related comment – throughout the manuscript the numbers of animals per experiment are provided, but it was never quite clear in each experiment how many worms, which experiments, etc. were used. This must be clarified well enough so the readers don’t have to dig around or guess.
- Figure 3 was confusing. What exactly is shown in 3a? Individuals? Averages? A subset of experiments? This must be clearly specified. The inset, which belongs either in a separate panel or in the supplement, made it only more confusing. Comparison of different standard models (R2 and m.a.e.) overcomplicated this already complicated panel. Figure legend did not help much.
- Line 176 promised to determine “which pathologies contribute the most to lifespan”. First, it shows essentially no correlation to any of the selected traits (fig 3c) and then an analysis in fig 3d that at least appears to be inconsistent with 3c.
- Line 187 “This analysis reveals that pharyngeal pathology is more predictive of lifespan in shorter-lived animals than in longer-lived ones (Fig. 3e).” How does this panel show this? Why are “long lived” animals in the legend of Fig. 3 >30 days, not 18 as above in the text of the paper?
- Discussion is excessive, speculative, and only somewhat connected with the findings reported in this study. In parts, it reads like a review or a polemic, not a discussion. Consider Figure 5. The only novel feature, and the only feature related to this article, is the repeat of Figure 1c. Plots of fertility and lifespan are very well established. The division of the lifespan into Stages 1 and 2 has also been made previously. What are we to learn from this figure? The last sentence of the legend trails off

Other:

- Is the title fair? Does ML really predict “underlying causes of death”?
- As far as I can see, the line re “high degree of accuracy (79%).” in the abstract derives from line 171. If so, it seems suboptimal to replace the measure of how well the model accounts for variability with the notion of accuracy.
- Several papers from Stuart Kim’s lab seemed appropriate to cite, for example, PMID: PMC3077391 and PMID: PMC4676719.
- In line 90, did you mean “Supplemental Fig. 4f-j”? And in line 140, did you mean 2e?

- I did not follow the logic of pairing uterine tumors and body bends in Fig. 2a. Panels 2b and 2c made intuitive sense, but this one did not. It became clearer following Fig 2d, although even that connection could probably be explained better. Making Fig 2d the first panel in the figure and re-ordering the narrative may help.
- In line 347, “A burst” seems too strong. Is it used to suggest that the gain in pathology is not linear? But why should it be? After all, since pathology scores asymptotically approach 100%, increase is likely to be greater in the early phase of aging. It’d be surprising otherwise.
- What exactly would qualify for a uterine tumor in males? Either exclude or modify fig. 4c.

Version 1:

Reviewer comments:

Reviewer #1

(Remarks to the Author)

Thanks to the authors for this manuscript about *C.elegans* quantitative genetics of aging platform. All of my addresses have been completed, and I recommend this manuscript for publication soon.

Reviewer #2

(Remarks to the Author)

The authors have addressed all of my previous critiques and suggestions. I am particularly pleased that they have corrected the issue with the feature importance metrics, this is a great improvement

I have no further comments.

Reviewer #3

(Remarks to the Author)

The authors have addressed some of my comments with some rewriting. My overall sense is that the manuscript is clearer now, even if it is a matter of opinion whether “surge” is much of an improvement over “burst”, etc.

I think this study captured something. Unfortunately, I can’t quite tell what it is. The problem of pathology scores isn’t that they are arbitrary, but rather that depending on the shape of the distribution, they may be uninformative at some ages and informative at others. The readers are left guessing whether this is the case. Fig. 2d shows very high correlation between the five pathology traits and three measures of behavioral deterioration. In contrast, Fig. 3c suggests very low correlation between the five pathology traits and lifespan. Is this due to effectively no correlation between behavioral deterioration and lifespan? Are the five pathology scores essentially independent of each other (since each alone is a poor lifespan predictor, but collectively they are good)? But that appears to contradict statements re coordinated onset of pathologies and high correlations shown in Fig. 2d. Finally, prediction quality is modest though Day 9 but picks up considerably by Day 14 and thereafter (Fig. 3a). Is this because dying or very nearly dying worms show dramatic, unmistakable pathology? This brings me back to the issue of distributions of pathology scores at each of sampled time points. Some of these concerns may be assuaged by presenting all collected data in a transparent way.

Having said the above, I see my role as strictly advisory – I pointed out what appeared to me to be concerning, less well substantiated or unclear points in the study. It is up to the authors to decide what to do with these comments.

Author responses to reviewer comments

Ref.: Ms. No. COMMSBIO-25-1850-T

Machine learning predicts lifespan and underlying causes of death in aging *C. elegans*

Reviewer #1 (Remarks to the Author):

In this manuscript #COMMSBIO-25-1850-T by Carina C. Kern et al. entitled 'Machine learning predicts lifespan and underlying causes of death in aging *C. elegans*', the authors describe the development and use of a new machine learning model to question the theoretical prediction that age-related pathologies are context-dependent.

The authors propose to study how interventions altering lifespan affect pathology, and how pathology affects lifespan.

To do this, the authors selected variables for evaluating the senescence process in the *C. elegans* model from observations taken from the scientific literature, as well as a large volume of observations presented in the manuscript.

By using direct observations with Nomarski microscopy, the authors selected the following as markers of senescence: pharynx, atrophy of the intestine, accumulation of yolk-rich pseudocoelomic lipoprotein pools, atrophy and fragmentation of the distal gonad, and development of teratoma-like uterine tumours. By establishing scores from 1 to 5 for each of the age-related pathology markers, the authors were able to train the machine learning model with up to 434 observations of wild nematodes as well as different mutant lines in combination with functional integrity markers of muscle (body bends, pharyngeal pumping) and intestine (atrophy and defecation).

The observations thus describe the impact of mutants in the Daf-2/Daf-16 aging regulatory pathway, Hsf-1, mTOR and the mitochondrial respiratory chain on the degree of severity of age-related pathology markers.

Tissue rescue experiments specific to DAF-16 function allow the authors to discuss the hierarchy of tumours observed in the control of senescence.

The impact of axenic feeding on the longevity of *C. elegans* is very well documented and has been evaluated in this work, providing excellent complementary results that provide a broader picture of explanations for the effects observed on the age-related pathology markers of the digestive apparatus. A clear negative correlation between age-related pathology markers occurrence and lifespan is described. Then, the authors ask if the observed age-related pathologies can appear independently of each other or in a dependent manner and propose, in fine, that pharyngeal deterioration plays a more important role than the other age-related pathology markers.

A comparative analysis of age-related pathology markers between the two sexes of *C. elegans* enabled the authors to clarify the different evolution of senescence and the impact of certain specific tissues in the process.

Finally, using this new model based on five-related pathologies to predict lifespan with an accuracy of 79%, the authors propose a two-stage model of worm hermaphrodites.

In this manuscript, the authors propose an innovative approach in an attempt to better define the order of events and the main tissues responsible during the senescence process in the *C. elegans* model.

This well-balanced manuscript represents the sum of high-quality experiments that have been carefully conducted. The results are very well discussed and benefit from a very critical approach that allows the reader to situate himself in the experimental strategy to answer the questions addressed.

The observation that this new tool can already predict senescence in this model up to a degree of 79% is highly original and, as proposed in the discussion section, points to the possibility of future improvements, such as taking into consideration other age-related pathology markers like sarcopenia, which can also be observed in this model. The statistical analyses, the number of participants and the experimental descriptions are described very precisely and are fully accessible to the experimenter.

I recommend the publication of this manuscript subject to the minor modifications as listed below:

Author response: We thank the reviewer for their constructive feedback, which has helped improve the overall quality of the manuscript.

In the Supplementary section of the manuscript

Supplementary fig.1a shows day 1 and day 14. To what age in days do the score 2, 3 and 4 images correspond?

Please indicate this information on fig.1a.

Author response: Done. Approximate age ranges at which the pathology of the given scores appear are now indicated.

In the references section of the manuscript

Names appearing in reference 17 in the references section of the manuscript must be corrected to appear with a capital letter at the beginning and in lower case thereafter.

Author response: This has been corrected.

The characters '<sc>' and '</sc>' in reference 38 should be deleted.

Author response: This has been done.

The title of the article in reference 51 should be corrected to lower case.

Author response: This has been corrected.

The names of the authors in capital letters in reference 54 must be corrected.

Author response: Done.

Reference 61 only includes the title of the article?

Author response: The reference has been completed.

Names appearing in reference 63 in the references section of the manuscript must be corrected to appear in upper case at the beginning and in lower case thereafter.

Author response: This has been corrected.

Names appearing in reference 70 in the references section of the manuscript must be corrected to appear in upper case at the beginning and in lower case thereafter.

Author response: This has been corrected.

Reviewer #2 (Remarks to the Author):

This manuscript describes the development of a random forest model that predicts nematode worm life-span based on measurements of 5 age-related pathologies measured at Day 11. Based on the features

that contributed most to predictions the authors suggest that diseases of the pharynx and intestines likely contribute to death in elderly *C. elegans*.

The manuscript is clearly and succinctly written, making it easy to review.

Overall, the methods used are methodologically and statistically sound, and the results are novel and interesting. That being said, I do have minor comments regarding terminologies, figure presentation and methods.

Author response: We thank the reviewer for their constructive feedback, which has helped improve the overall quality of the manuscript.

1. Although the text is nicely written, clear and concise, some of the figures could be improved. Particularly Fig2d needs resizing so the numbers and labels are visible. Y-axis labels are absent in some figures, and it is not always clear what the error bars represent (eg Fig1C).

Author response: Several of the figures have been revised to improve resolution and clarity, including Fig. 2d, and a Y axis label has been added to Fig. 2b, c. Error bars in Fig. 1C show the S.E.M., and this is now stated in the legend. Figure 2d has been revised.

2. The definition/public understanding of the word 'AI' has changed somewhat over recent years, and varies depending on who you ask. I personally, would not call these methods AI, and would prefer it be called only Machine Learning. I understand the temptation to refer to this as AI, since AI is a hot topic, but this paper is already great and does not need pumping-up with buzz words.

Author response: We are happy to refer solely to Machine Learning, and now do so.

3. Please double check how you calculated your feature importance scores in Figure 3e, I am concerned a mistake has been made. In figure 3e you show the mean importance metrics for each feature over the 20 'model runs' which I assume means 20 train:test data splits? Your plots contain error bars showing the s.e.m over those 20 splits, however, each run is so similar that the error bar is not visible, suggesting the importance metrics are identical across all 20 runs. Generally, splitting the data will have some impact on feature importance. It seems unusual for each split to be identical. Please double check your code for how you calculated these importance scores. If your scores are identical in all 20 splits it could suggest a problem in your code. If your procedure is correct, please show these results in a way that shows the importance metric for each run separately, like a jitter plot. (i.e. show 20 small dots for feature to better visualize how the importance varies across the 20 data splits).

Author response: In response to this we have rechecked the scripts, and indeed, in the initial submission, the feature importances were calculated for one train/test split only (hence the s.e.m was zero). We have computed the feature importances for all 20 random train/test splits (new Supplementary Fig. 8c, shown here).

4. In your section 'Use of ML analysis: insights and limitations' please mention that feature importance in ML models does not prove causation, it just indicates predictive ability. To show causation, future analysis will need to include 'causal inference' methods.

Author response: This is right, of course. The following has been added to the section specified. “It should be emphasized, however, that feature importance in ML models does not prove causation, but rather is an indicator of predictive ability; to demonstrate causation will require future direct tests of effects of pathology on late-life mortality.”

Reviewer #3 (Remarks to the Author):

Several figure panels were so pixilated that they were essentially unreadable. Not sure whether this was due to a conversion issue or to the original files, but it made it challenging to follow arguments.

Kern et al. presented the use of machine learning to predict lifespan based on scores of several aging-associated pathologies in *C. elegans* with help from other species. Overall, it is a laudable effort, but I have substantial concerns about how the results are presented (text and figures). The paper was not easy to follow. Results were presented in a telegraphic fashion. In several places I was not readily able to understand which data were used in the analysis, what assumptions were made, etc. For a study that reports a complicated set of ML models, this is a serious problem that makes it virtually impossible to evaluate central claims. This interesting and useful study is almost inaccessible. Please rewrite it!

We are grateful for the reviewer’s thoughtful comments, which have helped to improve the manuscript. We have now reworked the descriptions of the ML analysis to make it clearer exactly what was done, and the reasoning involved, as described below, and hope that our efforts are sufficient.

Major:

- Line 105/106. The statement ending with “largely self-evident, hence their description as pathological” struck me as underdeveloped. Do senescent pathologies cause death or are they byproducts of processes that cause death? Are they actually pathological? These distinctions are hard to make, but they are not trivial. It would not be reasonable to ask the authors to solve this problem, but a clear acknowledgement is necessary.

Author response: How to define the medical term “pathology” in biological terms is an interesting challenge. We now provide a more developed statement of meaning here, as follows.

“Senescence involves diverse changes that are pathological insofar as they disrupt biological function and, in some cases, contribute to late-life mortality. The age-related changes in anatomy documented here are clearly degenerative (organ atrophy, hypertrophy and structural deterioration) which strongly suggests that they are pathological in nature^{15,22}.”

The authors almost imply that they will present some evidence to back up their “self-evident” assertion, but in the page that follows (p.5) they do nothing of the sort. I don’t propose any specific experiments to substantiate the pathology claims, and the ability of the models to predict lifespan from pathology scores shows that they carry some relevant information, but I think it would serve the authors well to present a more seriously grounded argument here.

Author response: This is correct: there was a conclusion missing in this section. It has now been added as follows. “These two examples are at least consistent with the view that intestinal atrophy and uterine tumor formation are pathological processes.”

- The analysis described in line 127 and thereafter may be problematic. Although it may be valuable to compare progression of different pathologies, I am not convinced that the scheme proposed here would necessarily do so. First, for each trait an arbitrary scoring scheme was used. This is not to say that the

chosen categories were illogical or wrong. Rather, no information was provided whether a step up from score 2 to score 3 was a “similar gain in pathology” for Traits A and B.

Author response: This is an important point. Scoring in our study was based either on observed severity of degeneration or on a continuous variable - for example, change in intestinal mass. This approach is similar to the way that many human pathologies are assessed. We agree that it represents an approximation and is inherently subject to some degree of error.

Given the complexity of biological systems, it is currently not feasible to determine the full systemic effects of a specific pathology level. As such, it remains unclear whether a progression from, for instance, score 2 to score 3 reflects a proportional increase in pathological burden.

That said, we reason this does not undermine the utility of the scoring system, as the categories are grounded in observable differences in pathology severity. To address this more explicitly, we have clarified in the text that the scoring is an approximation. Additionally, we have also emphasized that this is precisely why the machine learning (ML) analysis is an appropriate tool - to uncover patterns that might not be evident through traditional observational methods.

Moreover, even if the scales were perfectly constructed, distributions of pathology scores among individuals could be considerably different for Traits C and D. This would limit the value of using Z scores.

Author response: We have data on distributions of traits. Note that pathology development in *C. elegans* is highly synchronous, as shown by our previous work, such that almost all animals have the highest severity of pathology development by day 14, with only marginal changes thereafter (see Figure 1c standard deviations).

Neither of these two major problems was seriously addressed in the manuscript, rendering Figure 2e at least unsubstantiated and more likely uninterpretable. One of several reasons to be concerned is that the overall picture of multiple traits may be largely driven by one or two traits because their variability and the scoring scheme made them the dominant drivers of the overall score. The authors may be better served by omitting this line of argument altogether.

Author response: We have edited the text to reflect that this is only an approximation with caveats, and that ML use was appropriate under the circumstances.

- Line 156. “randomly split the data into two groups: (i) one group of 80%”. REALLY unclear! What exactly do you mean by “the data” here? What does 80% refer to? A subset of all experiments? A subset of animals from each of experiments? Here and elsewhere, providing specific descriptions is essential.

Author response: We have enlarged this account to improve clarity. The figure legend now reads as follows.

“80% of the animal samples were used to train the model, and the remaining 20% used to evaluate the model. A sample or a data point corresponds to either a single animal ($n = 55$ wild-type individuals) or the mean value of a population subjected to a treatment/genotype ($n = 45$ population means) observed at different intervals (see Methods).”

- A related comment – throughout the manuscript the numbers of animals per experiment are provided, but it was never quite clear in each experiment how many worms, which experiments, etc. were used. This must be clarified well enough so the readers don’t have to dig around or guess.

Author response: This has now been clarified in the Figure legend as follows.

“The data comprises 438 samples \times 5 pathology features observed corresponding to either a single animal ($n = 55$ wild-type individuals) or the mean value of a population subjected to a treatment/genotype ($n = 45$ population-means) observed at different intervals (days 1,4,7,9,11,14).”

- Figure 3 was confusing. What exactly is shown in 3a? Individuals? Averages? A subset of experiments? This must be clearly specified. The inset, which belongs either in a separate panel or in the supplement, made it only more confusing. Comparison of different standard models (R2 and m.a.e.) overcomplicated this already complicated panel. Figure legend did not help much.

Author response: This has now been rewritten to improve clarity, as follows.

“Fig. 3 | Machine learning can predict lifespan from age-related pathology. a Selection of a random forest (RF) machine learning (ML) algorithm as the best predictor of lifespan from pathology, and use of the RF model to predict lifespan with $R^2 = 0.79$. **Left:** The RF model was identified as the best predictor of lifespan. R^2 values and mean absolute error (in days) are shown for different ML models, including linear regression (Linreg), Ridge regression, ElasticNet regression, Support Vector Machine (SVM), RF, and multilayer perceptron (MLP). All pathology measurements conducted up to day 14 of *C. elegans* lifespan were used. Day 14 was chosen because it is the time point at which maximum pathology severity was observed across all animals (see Figure 1c). **Right:** Lifespan prediction using the RF model achieved $R^2 = 0.79$. This analysis was repeated using the RF model, but instead of using all data up to day 14, data from up to various earlier time points (i.e. up to day 1, 4, 7, 9, 11, and 14 of adulthood) were used separately. The R^2 for each is shown in the inset. Day 11 outperformed all other time points, which is expected, as it represents the time point where pathology severity is generally highest before plateauing at its maximum around day 14 (see Figure 1c). The scatter dot plot shows data only up to day 11 of *C. elegans* lifespan. Each dot represents a held-out test sample and shows observed lifespan versus lifespan predicted by the RF model using pathology data. Eighty percent of the animal samples were used to train the model, and the remaining 20% were used to evaluate it. Each sample corresponds either to a single animal ($n = 55$ wild-type individuals) or to the mean value of a population subjected to a specific treatment or genotype ($n = 45$ population means) observed at different intervals (see Methods). For comparison of different models using only data up to day 11 of adulthood, see Supplementary Figure 8b. For coefficients of the linear regression model, see Supplementary Figure 8b. p -values compare the RF model to other models.”

- Line 176 promised to determine “which pathologies contribute the most to lifespan”. First, it shows essentially no correlation to any of the selected traits (fig 3c) and then an analysis in fig 3d that at least appears to be inconsistent with 3c.

Author response: This has now been rewritten to improve clarity, as follows.

“The apparent discrepancy arises because panel 3c reports linear correlations on the raw features, while panel 3d shows Random-Forest “mean decrease in impurity” (MDI), which captures non-linear effects and interactions among features. A feature can therefore have a low linear correlation with lifespan yet still be important in the multivariate, non-linear RF model. This clarification was added as a footnote when discussing the figure in the results section.”

- Line 187 “This analysis reveals that pharyngeal pathology is more predictive of lifespan in shorter-lived animals than in longer-lived ones (Fig. 3e).” How does this panel show this? Why are “long lived” animals in the legend of Fig. 3 >30 days, not 18 as above in the text of the paper?

Author response: This error has been corrected; yes, 18 is the correct value.

- Discussion is excessive, speculative, and only somewhat connected with the findings reported in this study. In parts, it reads like a review or a polemic, not a discussion. Consider Figure 5. The only novel feature, and the only feature related to this article, is the repeat of Figure 1c. Plots of fertility and

lifespan are very well established. The division of the lifespan into Stages 1 and 2 has also been made previously. What are we to learn from this figure? The last sentence of the legend trails off

Author response: After re-examining the discussion in the light of these comments, we take it that they refer mainly to the final section of the discussion, including Figure 5. This section was included in an effort to add value and interest to the article, but the reviewer is correct that it is relatively superfluous, so it has been removed.

Other:

- Is the title fair? Does ML really predict “underlying causes of death”?

Author response: This is a fair point. Taken alone, the word “predict” does not specify any particular level of confidence, hence its use in the title is somewhat vague. We have therefore modified the title to read “Machine learning predicts lifespan and suggests underlying causes of death in aging *C. elegans*”.

- As far as I can see, the line re “high degree of accuracy (79%).” in the abstract derives from line 171. If so, it seems suboptimal to replace the measure of how well the model accounts for variability with the notion of accuracy.

Author response: Agreed. This is changed to variance.

- Several papers from Stuart Kim’s lab seemed appropriate to cite, for example, PMID: PMC3077391 and PMID: PMC4676719.

Author response: We are familiar with these studies, whose topics are related to, though somewhat orthogonal to, the present study. The earlier work (Sanchez-Blanco and Kim, 2011) relates to bacterial pathogenicity and causes of stochastic variation in lifespan rather than, as in the present study, endogenous senescent pathologies that contribute to late-life mortality. However the second study (Zimmerman et al., 2015) is germane to findings relating to uterine tumor development and mortality, and we now refer it as follows.

“This could imply that amelioration of pathologies that limit life leads to their replacement by new causes of late-life mortality, here bacterial infection and uterine tumor development, respectively. In the latter case, a possibility is that life-shortening secreted uterine proteins play a role (Zimmerman et al., 2015)”

- In line 90, did you mean “Supplemental Fig. 4f-j”? And in line 140, did you mean 2e?

Author response: In both cases yes. These errors have now been corrected.

- I did not follow the logic of pairing uterine tumors and body bends in Fig. 2a. Panels 2b and 2c made intuitive sense, but this one did not. It became clearer following Fig 2d, although even that connection could probably be explained better. Making Fig 2d the first panel in the figure and re-ordering the narrative may help.

Author response: To improve clarity, the pairing of intestinal atrophy and defecation is now presented before the uterine tumor example. The hypothesis about how uterine tumors might affect body bends is now explained as follows.

“Tumors frequently attain a great size, such that they fill much of the body cavity in the mid-body region; one possibility is that this impedes body bends, either by increasing mid-body rigidity, or due to the tumor pressing against body wall muscles.”

Making Fig. 2d the first panel is a good idea in principle, and we attempted to make this change, but in practice it created difficulties in terms of the graphic organization of the figure.

- In line 347, “A burst” seems too strong. Is it used to suggest that the gain in pathology is not linear? But why should it be? After all, since pathology scores asymptotically approach 100%, increase is likely to be greater in the early phase of aging. It’d be surprising otherwise.

Author response: “Burst” here is intended to denote a relatively brief period of rapid development of pathology, mainly between day 4 and 8 of adulthood. It is plausible that the rate of pathology development slows after this period, though some of the apparent deceleration may reflect inter-individual variability. We agree that “burst” is rather too strong, and now use the term “surge” instead.

- What exactly would qualify for a uterine tumor in males? Either exclude or modify fig. 4c.

Author response: This error has been corrected. Fig. 4c is now labelled “Germline tumor”.

Sanchez-Blanco, A. and Kim, S.K., 2011. Variable pathogenicity determines individual lifespan in *Caenorhabditis elegans*. *PLoS Genet.* 7, e1002047.

Zimmerman, S., Hinkson, I., Elias, J. and Kim, S., 2015. Reproductive aging drives protein accumulation in the uterus and limits lifespan in *C. elegans*. *PLoS Genet.* 11, e1005725.

Response to Reviewers

We thank all three reviewers for their constructive comments and suggestions. We are pleased that Reviewers #1 and #2 are fully supportive of the revised manuscript, and we are grateful to Reviewer #3 for raising thoughtful questions that allow us to further clarify our work. Below we address Reviewer #3's points in detail.

Reviewer #1

Comment: All addresses have been completed, and I recommend this manuscript for publication soon.

Response: We thank the reviewer for the positive assessment and support for publication.

Reviewer #2

Comment: The authors have addressed all of my previous critiques and suggestions. I am particularly pleased that they have corrected the issue with the feature importance metrics, this is a great improvement. I have no further comments.

Response: We thank the reviewer for their helpful input throughout the review process, and for their supportive comments on our revisions.

Reviewer #3

Comment:

The problem of pathology scores isn't that they are arbitrary, but rather that depending on the shape of the distribution, they may be uninformative at some ages and informative at others. The readers are left guessing whether this is the case.

Response:

We agree that the informativeness of pathology scores varies with age, and that score for different pathologies can be inter-dependent. To clarify, our approach models pathology progression over time, rather than static values at single ages. Significantly, the ML model supported predictions of the most informative ages, based on the observation of pathology progression through time (see Figure 1c). As seen in Figure 1c, pathology levels begin to plateau after Day 11 of adulthood, i.e. maximum pathology severity is being reached (e.g. maximum atrophy of the gut has occurred etc). Thus, the prediction is that measuring pathology progression until Day 11 will be the most informative. Indeed, as shown in Fig. 3a, the random forest model (which is also based on changes across time points, rather than absolute pathology scores alone) achieved the highest predictive power using data points up to Day 11; predictive accuracy is modest at earlier time points but increases sharply by Day 11, and post-Day 11 predictive power is lower. We have clarified this in the Discussion, 249-252.

Comment:

Fig. 2d shows very high correlation between the five pathology traits and three measures of behavioral deterioration. In contrast, Fig. 3c suggests very low correlation between the five pathology traits and lifespan. Is this due to effectively no correlation between behavioral deterioration and lifespan?

Response: In this study we did not measure the correlation between behavioural decline and lifespan, however other studies have shown a positive, but weak, correlation between behavioural deterioration (reduced movement, pharyngeal pumping) lifespan in *C. elegans* (e.g. Huang et al. 2004; Podshivalova et al. 2017). Our findings are consistent with this: pathology measures correlate strongly with functional decline, but their relationship with mortality is more complex.

Comment:

Are the five pathology scores essentially independent of each other (since each alone is a poor lifespan predictor, but collectively they are good)? But that appears to contradict statements re coordinated onset of pathologies and high correlations shown in Fig. 2d.

Response: The correlation between scores for different pathologies is variable. As shown in Fig. 3c, correlation between pathologies ranges from low (Pearson correlation coefficient $r = 0.25$) to relatively high (Pearson correlation coefficient $r = 0.81$), suggesting some pathologies progress in a coordinated manner. This could explain why individual scores can be weak predictors of lifespan, yet collectively provide strong predictive power captured by the random forest model. To clarify: the onset of the five major pathologies tends to occur in a coordinated manner in early to mid-adulthood, as shown in Fig. 1c. However, the subsequent development of each pathology is not necessarily closely correlated with the others across different conditions. The correlations between individual pathologies are shown in Fig. 3c rather than in Fig. 2d.

Comment:

Finally, prediction quality is modest through Day 9 but picks up considerably by Day 14 and thereafter (Fig. 3a). Is this because dying or very nearly dying worms show dramatic, unmistakable pathology?

Response: This is an interesting point. As the reviewer points out, prediction increases sharply by Day 11 (rather than Day 14), however we do not think this reflects that animals are already dying or nearly dead. At 20 °C, the mean lifespan of wild-type *C. elegans* in our dataset is ~18 days, so Day 11 corresponds to mid-life rather than late-life. In the combined dataset we used for ML analysis, lifespans across different treatments and genotypes ranged from 7.5 to 40 days (Fig. 3b). In spite of the high diversity in lifespan of these populations, the RF model achieved high predictive accuracy, indicating that the model is not just capturing terminal pathology in moribund worms. It is, therefore, more likely that the greater predictive power arises from the fact that later age provides both more data and more time for animals with delayed pathology progression to exhibit differences in their disease trajectory.

Comment:

Having said the above, I see my role as strictly advisory – I pointed out what appeared to me to be concerning, less well substantiated or unclear points in the study. It is up to the authors to decide what to do with these comments.

Response: We thank the reviewer for taking time to review our manuscript, and appreciate the input. We hope we have clarified the points raised.